# From Fathers to Fathers—Telephone-Based Peer Support: A Feasibility Study

Ewa Andersson * , Lisa Espinosa and Michael B. Wells

Reproductive Health, Department of Women's and Children's Health, Karolinska Institutet, 17177 Stockholm, Sweden; lisa.espinosa@gmail.com (L.E.); michael.wells@ki.se (M.B.W.)
* Correspondence: ewa.andersson@ki.se

**Abstract:** Background: Men can struggle with adapting to their new roles as they transition into fatherhood. While social support has been shown to be effective at aiding this transition, little research has focused on the implementation of, and satisfaction with, telephone-based peer support for new fathers. Aims: This qualitative study aimed to investigate the implementation of, and satisfaction with, a telephone-based peer support program for new fathers. Methods: A qualitative study with 13 interviews of first-time fathers and peers was analysed using content analysis, in accordance with Elo and Kyngäs. Individual interviews were conducted using a semi-structured interview guide that lasted between 30–45 min. Results: Two themes emerged from the fathers' interviews (n = 6): (1) conditions that affect the telephone support experience; and (2) the importance of support. The fathers appreciated the confirmation stories shared by their peers, as these stories served as valuable examples that they could adapt and incorporate into their own parenting approaches. Two themes emerged from the peer interviews (n = 7): (1) peers' own role and experience; and (2) the Importance of listening to fathers. Peers felt appreciated and acted like role models for new fathers, helping them to adjust to parenting life. Study limitations: The results may not transfer to multi-time fathers. Conclusions: The findings of this study provide valuable insights into the potential benefits and challenges of implementing a telephone-based peer support program for first-time fathers, which could further inform similar interventions.

**Keywords:** adherence; feasibility; first-time fathers; interview; telephone peer support; qualitative study

## 1. Background

Transitioning to parenthood leads to physical, psychological and social changes in the individual (Sanchez and Thomson 1997). Some fathers describe this transition as an emotional rollercoaster (Halpern and Perry-Jenkins 2016) that can negatively impact their well-being (Knoester and Eggebeen 2006) and/or their relationship with their partner (Figueiredo et al. 2008). In addition, poor paternal mental health negatively affects the mental and physical health of children (Tichovolsky et al. 2018; Wolicki et al. 2021), and has been found to predict higher risk of child behavioral problems, conduct disorders and peer relationship difficulties (Ramchandani et al. 2008). Receiving clinical professional support, such as from midwives and/or child health nurses, during the transition to fatherhood has been shown to be associated with a reduction in fathers' depressive symptoms (Wells and Aronson 2021). However, fathers often feel marginalized by perinatal healthcare (Fletcher et al. 2006), and therefore do not always receive the support they would like from clinical professionals (Shorey et al. 2017; Wells et al. 2017). This support gap in health care professional assistance contributes to fathers themselves lacking knowledge about their risk of developing postpartum depression (Nazareth 2011), and increases their risk of experiencing parental stress due to social isolation (Skreden et al. 2012). A recent scoping review, in seeking to better support fathers, concluded that fathers would benefit from the

support of peers, such as another father, and familial support, as this would enable them to better adjust to their new parental role (Leahy-Warren et al. 2023). Non-clinical avenues of support may also be further warranted to ensure that fathers have a healthy transition into their new parental role.

## 2. Peer Support to Improve Wellbeing

Social support is defined as the provision or exchange of emotional, informational and/or instrumental resources (Cohen and Wills 1985). A lack of social support is considered a major health risk, comparable to smoking or obesity (Gruenewald et al. 2009; House et al. 2004). One common way of receiving social support is through peers (Simoni et al. 2011). Peer support is theoretically embedded within the concept of social support and is a beneficial way of promoting health across a variety of populations (Dennis 2003). Peers with similar backgrounds can understand a target population's situation in a way that family, friends or healthcare professionals may not (Dennis et al. 2009). Consequently, the provision of peer support usually needs to extend beyond a natural or embedded support network into a newly created network where experientially similar "others" can be found (Wells and Aronson 2021; Dennis 2010).

Within a healthcare context, peer support is defined as a multifaceted form of social support that incorporates the provision of (a) informational support, such as knowledge, facts, and suggestions; (b) emotional support, such as attentive listening, caring and reassurance; and (c) appraisal support, such as motivation, encouragement and positive communication, delivered by a person who has similar characteristics or health conditions (Dennis 2003). Similarly, Eriksson and Salzmann-Erikson (2013) concluded there are three main categories of online paternal peer support: (1) encouragement, which is similar to appraisal support; (2) advice, which is similar to informational support; and (3) confirmation, where peers shared their similar experiences of events and their handling of events with new fathers. These types of peer support directly protect an individual's physical and psychological well-being and indirectly foster a sense of control (Thoits 2011). Peers can also exert influence, including on the individual's coping strategies and sense of personal control (Taylor and Stanton 2007). As such, father-to-father peer support programs have found that after expectant fathers received information from fathers, they were better able to support their partner's breastfeeding efforts (Kim 2018). In addition, new mothers who received peer support had less risk of developing postpartum depression (Dennis et al. 2009).

## 3. Implementation Fidelity

Telephone-based peer support programs for new mothers has been regarded with high satisfaction by mothers, who have also noted it is highly feasible (Dennis 2010; Law et al. 2021). However, little is known about the efficacy of providing a similar peer support program for fathers. In the current pilot study, we developed and implemented a telephone-based peer support program for fathers during the first 16 postnatal weeks. The aim of the current study was to assess the implementation fidelity and satisfaction of this peer support program by using semi-structured qualitative interviews.

"Implementation fidelity" refers to adherence to the program, including the extent to which a program is implemented as intended. "Exposure/dose" refers to the program's frequency and duration. "Quality of delivery" discusses the implementation (how and in what way) of the program. "Participant responsiveness" includes participants' participation and estimation of the relevance of the program. And "program differentiation" refers to the essential components that help make the program successful (Law et al. 2021). Better understanding of the implementation fidelity can help researchers to interpret if the outcomes are due to the study design (Carroll et al. 2007; Law et al. 2021).

The transition to fatherhood is a complex and multifaceted experience that significantly impacts the lives of men. Despite a growing recognition of the importance of fathers in parenting (Grau et al. 2022), and professionals increasingly starting to provide support

to new fathers (Odonde et al. 2022) there is still a significant gap in the feasibility of universally-offered peer support for new fathers. Although many fathers are asking for peer support programs, relatively few peer interventions currently seem to exist (Leahy-Warren et al. 2023; Ocampo et al. 2021). The aim of this study is to explore fathers' and peers' experiences of a telephone-based peer support program for new fathers.

## 4. Methods

The current study uses a qualitative design, in acknowledgement of the significance of grasping the meanings that individuals attach to their experiences. This methodology acknowledges that reality is subjective and underscores the need to comprehend interpretations from individuals within distinct contexts. This process involves recognizing how fathers and peers perceive and interpret the same phenomenon, while acknowledging variations among individuals. Additionally, using a qualitative approach can provide insights that help to develop interventions and can assist the identification of obstacles and catalysts, and thereby contribute to successful implementation (Ramanadhan et al. 2021).

### 4.1. Study Design

4.1.1. The Telephone-Based Peer Support Program for Fathers

The telephone-based peer support program for new fathers (TPSP-F) was designed by two authors (removed for peer review) on the basis of Dennis' model of peer support for mothers (Dennis 2003). The TPSP-F involved three groups of people: (i) mentors, (ii) peers, and (iii) new fathers. Mentors (n = 3) were recruited from a non-profit organization called Män för jämställldhet (MÄN; men for gender equality) and consisted of professionals with 5 to 20 years of experience of leading father groups and supporting fathers as they transition into parenthood. Their primary role included being available by telephone when the peers had questions or wanted to raise concerns. Prior to starting the study, mentors received a half-day training from (removed for peer review) on how they could best support the peers. Mentors were an essential component of training and supported peers throughout the program.

Peers (n = 17) were experienced fathers with children aged between toddlerhood to adolescents, who volunteered their time to provide telephone-based peer support to fathers. These peers were trained by the mentors in two three-hour sessions. The training addressed three critical areas of peer support: (i) informational support, (ii) emotional support, and (iii) appraisal support (Dennis 2010; Simoni et al. 2011). Each area included guidance on, and discussion of, fathers' mental health, the father–child relationship and the coparenting relationship. During the training, mentors and peers discussed the content they might talk about with fathers and practiced how to talk with fathers while using motivational interviewing techniques (Erickson et al. 2005). Peers also learned how to document their phone conversations with fathers in a peer volunteer activity log, which was developed for the breastfeeding trial and modified for the pilot trial with fathers (Dennis 2002).

Fathers (n = 26) had to meet the inclusion criteria: (i) be in the last trimester or one month postnatal; (ii) aged 18–49 years-of-age; (iii) speak Swedish or English; and (iv) have a healthy baby. Although it was not a requirement, all fathers reported being in a heterosexual couple relationship and living with the mother of their child. Most fathers were also first-time fathers (n = 22), with a mean age of 32.1-years-of-age.

4.1.2. Frequency and Duration

Fathers gave explicit consent for the research team to share their email address with the randomly matched peers. Within a two-week period after the training, peers contacted the fathers (via email) and booked a first phone call. Text messages were then used to schedule phone calls. Fathers and peers themselves decided how many times they would talk but were instructed to talk for a minimum of once a month for four months. Three of the peers supported two fathers each, albeit not simultaneously. They were asked to discuss the father's feelings, including his mood, and his relationship with his child and

partner, including their romantic relationship and the coparenting relationship. Fathers were also asked if there was any topic they wished to discuss. On average, calls lasted 60 min. Although peers were instructed to contact the researchers if they believed any father needed further psychological support from a professional (e.g., in instances self-harming or making suicidal statements), no such need was reported.

### 4.2. Participant Recruitment

Fathers and peers were recruited via advertisements on social media, posters at antenatal clinics and via our collaborations with the organizations MÄN and Mamma till Mamma (Mom to Mom). After the program ended, six first-time fathers and seven peers responded positively to the invitation to be interviewed (n = 13), and subsequently participated in the current study.

### 4.3. Data Collection

Data collection consisted of qualitative interviews of first-time fathers and peers who participated in the TPSP-F pilot study. All interviews were conducted in Swedish, as all participants were native Swedish speakers. The interviews were conducted via Zoom three weeks to two months after their last peer support during Spring 2021, by two researcher and two midwifery master's degree students.

In using a semi-structured interview guide (Supplementary File S1), four interviewers (EA, LE, IA, and CS) independently interviewed either a first-time father or a peer, following prior research by father groups (Wells et al. 2021) and a pilot test with one participant. The interviews focused on the experiences of giving support (peers) or receiving support (fathers). Additional questions were asked to prompt, clarify, and to encourage participants to expand their statements. Interviews lasted 30 to 45 min, and they were all recorded and later transcribed verbatim by EA, IA and CS. All transcripts were switched between the three transcriptionists to check for accuracy.

### 4.4. Data Analysis

Data was analyzed using content analysis, in accordance with Elo and Kyngäs (2008). EA, CS, IA independently read the transcripts and made an open coding (notes and headings are written in the text while reading it). All authors identified meaning units, followed by condensing, coding and abstraction into subcategories, to later classify the results into categories and themes. Researchers discussed their opinions before compiling the categories. All authors agreed on the final categories and themes, and then directly translated the results from the Swedish to the English language.

### 4.5. Ethics

The regional ethics board approved the peer support program and the interviews (Dnr:2015/1662-31 and 2016/217-32). All participants were informed before starting the program that they would later be asked to participate in an interview regarding their experiences and that their data would be protected according to the general data protection regulation (GDPR). All participants were given a consent form that detailed the purpose of the study, stated that their participation was completely voluntary, and reassured participants that they would remain anonymous after participating in the program.

## 5. Findings

First, we report the results from the first-time fathers (n = 6), followed by the peers' (n = 7) results, which both independently produced two main themes (Tables 1 and 2). The fathers' themes included: (1) conditions that affect the telephone support experience; and (2) the importance of support; while peers' themes included: (1) their own role and experience and (2) reflections on the father needing support.

**Table 1.** Overview of main themes and categories that emerged from the analysis of interviews with fathers.

| Main Themes | Categories |
|---|---|
| Conditions that affect the experience of telephone support | Length of the conversation |
| | Interval and beginning the calls |
| | Timing of the phone calls |
| | The telephone support format |
| The importance of support | Fatherly Advice |
| | Life situation—a new phase in life |
| | A need for social support |

**Table 2.** Overview of main themes and categories that emerged from the analysis of interviews with peers.

| Main Themes | Categories |
|---|---|
| Peer's own role and experience | Reasons for being a peer |
| | Anonymous phone support and conversation |
| Importance of listening to fathers | Dare to talk about your feelings |
| | The fathers' need for social support |

*5.1. Fathers*

**First theme: Conditions that affect the telephone support experience**

This theme focused on the benefits and challenges of the program for first-time fathers and comprised four categories: (1) *length of the conversation*, (2) *interval and beginning of the calls*, (3) *timing of the phone calls* and (4) *the telephone support format*.

(1)   Length of the conversation

Before the actual phone call, first-time fathers spoke to their peer, and discussed and agreed with their peer how long they would converse. Regardless of the length of the call, fathers stated that the length of the peer support phone call was appropriate and that conversations over an hour would be too long. Fathers who had relatively short conversations, such as 20 min, reported that it would have been difficult to have a longer phone conversation due to childcare requirements, or their need to sleep or work. Fathers believed that a mix of texting and phone calls worked well.

> *We booked appointments by text and then we were called. He introduced himself, and it was obvious who we were when we SMS [texted] about follow-up times. It worked for me*—Father 2.

(2)   Interval and beginning the calls

The intervals between calls varied. Fathers reported having between four and six phone calls at an interval of two–three weeks at first, and then once a month afterwards. Fathers thought that this time interval was reasonable. Fathers reported that while continuing the support after the program ended was not necessary, it was clear to them that they could reach out to their peer, if the need arose. Occasionally, first-time fathers would have liked to have had a follow-up conversation with a peer some months later, just to provide an update on how parenting life was going.

> *I don't feel the need to talk more, but it would just be fun. But maybe after a longer period, it would be nice. It would have been good to follow-up*—Father 1.

One first-time father who started receiving support when his child was two-months-old stated that he wished he could have had support earlier, as he found the first period

after the birth to belife changing and had many thoughts and questions that peer support could have helped with.

(3)    Timing of the phone calls

Fathers reported that it was beneficial to have the peer support phone calls when alone (and therefore not distracted), such as during their commute to and from work or during stroller walks. They perceived this as a valuable use of time, as this was their own time when they could focus entirely on the conversation. By talking without distractions, fathers found the conversations deeper and more helpful. Importantly, fathers said they found it difficult to set aside private time for conversations, as they felt that they had too many other tasks, making it difficult to focus exclusively on the peer support phone conversations.

(4)    The telephone support format

Fathers reported that the anonymous format by phone was beneficial. By not seeing the other person, they felt the conversation became personal faster.

*Good thing about not seeing each other—it became a smaller threshold, because it can be judgmental on appearance*—Father 6.

Fathers however reported that meeting the peer could have been beneficial, especially for physical contact.

*In some situations, it would have been nice to sometimes have a hug or pat on the back*—Father 1.

Fathers initially reported they were somewhat worried that the conversations would be quiet, but said this fear quickly dissipated as conversations developed after the call started. However, fathers noted that it was up to them to talk to get the conversation going again. They suggested peers could have a protocol, including different themes that could be covered in each conversation, to help encourage further dialogues.

**Second theme: The importance of support**

This theme included three categories: (1) fatherly advice, (2) life situation–a new life phase; and (3) a need for social support. In engaging with this theme, fathers spoke about their experiences of talking with the peer, and explained why it was important.

(1)    Fatherly Advice

Fathers reported that it was nice to have someone to talk with and be supported in their new parental role, and also thought that the peers were competent at helping them with any of their parenting questions and concerns. They appreciated that the peers were fathers themselves. Fathers felt that the peers' own confirmation stories were valuable and enabled them to adapt and incorporate those stories into their own parenting.

*This bit is kind of an advice, but not directly—I appreciated it, the stories*—Fathers 4

Fathers greatly appreciated that the peers actively listened, asked additional questions in seeking clarification, provided advice and parenting tips, and confirmed their thoughts and concerns. These additional questions were perceived as peers being interested and caring about the fathers' adjustment into parenthood. Fathers also thought it was nice to be able to talk to someone who was independent and non-judgmental.

The need for peer support differed amongst the fathers. For example, some fathers wanted to vent their thoughts and concerns about being a good father and partner, while others wanted practical advice. Fathers felt that the program provided them with confirmation about their new role.

*Here he's also been very supportive and affirmative in that 'You're a good dad, you're a good man, and you're good at this.*—Father 3

(2)    Life situation—a new life phase

In reflecting on talking with peers, fathers reported reflecting on their parental role and recognized that being a first-time father was an important and large transition in their

life. They noted differences in family dynamics, as they transitioned from being a dyadic couple to a triadic family. One father described raising an infant as a 24-h job. As such, fathers saw the telephone support as an opportunity to have an ongoing dialogue with someone about what it was like to be a father. For example, one father did not know how to deal with his sad child at first but was encouraged by the peer.

> *I'm becoming more confident in my own [parental] role. I don't care about the baby screaming. I'm calm and that's perfectly fine.*—Father 4

(3) A need for social support

Fathers said they did not have any friends or family that they could to talk to about becoming a dad. They noted that this was because they were the first to have children and thought that their non-parent friends would find it difficult to relate to being a parent when they were not one themselves. Fathers said that they did not have large social networks, and thus had quite limited options for receiving support—their only source of parenting support came from their partner, who was often a first-time parent themselves.

> *What was nice about having a peer father was that I'm the first in my group of friends to become a parent and automatically didn't have a natural contact to turn to.*—Father 2

Although some fathers had already asked for and received professional help from psychologists, others mentioned that men generally find it difficult to talk about their feelings or ask for help. As such, one father expressed that receiving support from an anonymous person was a good way to reach fathers who need it, without the worry of being judged.

In addition, fathers appreciated that peers were only calling to focus on them and their parenting needs. They viewed the peer as an objective person who could relate to their situation, who they could talk with. Fathers appreciated that this peer program emphasized "support".

> *It felt good that it was called support and not help. It felt more like, here is a person who went through something similar*—Father 1.

Fathers felt that they were strengthened in their parental role thanks to the contact with their peer.

> *The confirmation in the talk that you are a good father [...], I feel much stronger and strengthened as a father now than I did before my conversations with my peer*—Father 3.

*5.2. The Peers*

**First theme: Peers' own role and experience**

This theme described the peers' own experiences of becoming fathers and what support they needed, leading them to reflection on their current peer role, which comprised two categories: (1) *reasons for being a peer* and (2) *anonymous phone support and conversations.*

*(1) Reasons for being a peer*

Peers were motivated to participate because they believed that there were benefits to providing individual-based support, such as confirming to fathers that they were on the right track and that their voices were heard. Peers believed that this would lead fathers to feel they were being understood. Peers felt secure in their parental role and wanted to share knowledge and be a role model for other fathers. Many peers also reported that they had previously led father groups.

> *I felt in the last year that I am a role model for many men and then this is a mission I would like to be part of and share*—Peer 4.

A peer, with no previous experience of leading father groups, agreed to participate because he felt that individual peer support was something he could have benefitted from himself.

*I felt that there were many questions, a lot of feelings within me, that I didn't dare talk to anyone about, and I was unsure. And it would have been nice to have someone who. . . could unload and talk to someone and gain perspective*—Peer 5.

Peers unanimously agreed that this peer program had benefited them. One peer expressed that it was a privilege to help another person's life, while another expressed that peer assistance provided important and necessary support to fathers.

*(2)  Anonymous phone support and conversation*

Peers expressed positive feelings toward the phone-based (vs. in-person) support and thought it was a good way to provide support. Furthermore, peers liked that the program was anonymous. They believed by creating a safe space to share issues, especially on taboo subjects, anonymity could make fathers feel less vulnerable, especially in a relatively short period of time.

*It [the program being anonymous] was an incredible benefit for the person I was the contact for because they didn't know me. [. . .] I don't know anything about the person, have no relatives, no friends with that person, which means that that person can talk about things you might not talk to anyone else about*—Peer 7.

The peers tried to listen to and support fathers on all levels, including their role as a father, the father's mood, their romantic relationship, and their coparenting relationship. Peers let fathers share their own thoughts and concerns, and mostly listened, especially at the beginning of the conversations. Then, they would normally validate the fathers' feelings and thoughts before moving the dialogue further. They often provided parenting tips and talked about their own fathering experiences.

*We had a mirror-image relationship; he and his partner and me and my partner. There I could explain things from her perspective because I resembled her in relation to my partner.*—Peer 2

Peers expressed that they tried to challenge the fathers' thoughts or expectations, especially if the father was not considering other perspectives.

*. . . and then I came with some challenges to him: "Can you think like this? Can you think like that?", and so on. He was very happy about it.*—Peer 6

Peers believed the free format of the conversations was an advantage of the program. For example, when fathers had many topics to discuss, they could let fathers lead the dialogue and express themselves, which allowed the peers to have time to validate the father's experiences. In this way, the time belonged to the father, who could define the conversation. On the other hand, peers sometimes thought it would have been beneficial to have an agenda for each call, especially if the conversations led to a dead-end and were difficult to restart. Finally, peers felt that some fathers had difficulty finding time for the phone calls and tried to remain flexible to the fathers' time schedule.

**Second theme: Importance of listening to fathers**

This theme included the importance of creating a sensitive and respectful dialogue with fathers to be able to support them. It comprised two categories: (1) *dare to talk about your feelings*; and (2) *the fathers' need for individual support*.

(1)  Dare to talk about your feelings

Peers mentioned that men generally found it difficult to open up and talk about their feelings, and claimed on this basis that this type of support could be very meaningful and helpful. Peers believed that people always need additional support whenever they enter a new phase or take on a new role in life. They however acknowledged that father-to-father support systems are not common today.

*There aren't that many natural forums where men can receive support in a positive way. . . It's mostly beer and football that are in focus. . . These positive forums are not completely obvious, while I see this [peer support] as such.*—Peer 1

Peers believed that by participating in this program, they also enhanced their own parenting skills by improving their listening skills and ability to see issues from other people's perspectives. Thanks to these conversations, which gave them time to practice putting feelings into words, they stated that they became better at talking about feelings with their partner and friends.

(2)   The fathers' need for individual support

Peers felt that fathers mostly just wanted to talk and receive validation that their feelings and thoughts about parenting were being heard. Peers tried to encourage fathers to see the calls as an opportunity to take care of themselves, which they considered important. They advised fathers to take personal time out from their relationship and family life and instead focus on their own mental and physical well-being. They believed that by doing this, fathers could instead consider their own thoughts and feelings regarding the new roles, both in relation to their partner and as a family.

> *We (men) are so used to always thinking that we should take care of everything, but then when the child is born, some situations arise when we think that it is the woman who should decide here, as if we are satisfied that now she will finally have the rights to decide. But that means we don't, at all, reflect on "what do I really want myself".*—Peer 6

Peers acknowledged that they needed to listen and provide support on various issues. For example, they stated that fathers were experiencing actual challenges in redefining their romantic relationship, and learning how to coparent. One peer reported that fathers would sometimes create or embellish a problem just to have something to talk about, rather than seeing the phone call as a bonding moment or just an opportunity to converse with a like-minded peer.

The peers agreed that the received social support was a positive experience for fathers and for themselves. Peers expressed a willingness to continue talking with the father even after the program ended, if the father so desired, while others believed that the fathers did not require additional support. One peer who had supported two expectant fathers during the prenatal period stated that after the babies were born, both fathers felt secure in their role and did not need as much social support. Another peer also believed that social support should start in the prenatal period.

> *I felt that it was good to get in touch beforehand. Then you could talk about preparatory matters.*—Peer 3

## 6. Discussion

The current study interviewed both first-time fathers and peers regarding the implementation fidelity and satisfaction of a telephone peer support program during the first 16 postnatal weeks. Overall, fathers and peers believed that the program was beneficial and could aid fathers in their transition to parenthood. Furthermore, fathers stated that the delivery of the program by the peers was of high quality. All fathers completed the program, suggesting high participant responsiveness. As such, both groups adhered to the program's protocols. Fathers reported being helped in their parental development, and peers reported developing communication skills with other men. Fathers further expressed that, if it had not been for this peer support program (TPSP-F), they would have had nobody to talk about fatherhood issues.

The current TPSP-F study was inspired by the United Nations Convention on the Rights of the Child (United Nations 1989) as well as Swedish gender equality policies, such as those outlined in the Parenting Support Commission Report (Social Department 2008), which state that all members of the family should be included and supported (Eklund and Lundqvist 2021). However, even though there has been dramatic progress on supporting fathers professionally in Sweden, including by creating a new specific visit for fathers/non-birthing parents at Swedish child health centers (Rikshandboken 2019), fathers may not be invited to the visit or be unable to attend; in addition, nurses might not cover all of the information fathers need (Odonde et al. 2022). Thus, new fathers can still feel as though

they have received little attention, leaving them to feel unsupported and even invisible during the postpartum period (Edhborg et al. 2016), which can consequently impact their mental health (Wells and Aronson 2021) and coparenting relationship (Wells and Jeon 2023). Trying to invoke institutional change could be challenging, as there may be: (i) little perceived need to support fathers; (ii) inadequate testing of best practices to increase fathers' involvement (Bremberg 2016); and (iii) nurses can often feel and/or be overworked (Anskär et al. 2022); finding solutions outside of the medical field to further support fathers may therefore be necessary.

Peer support can however influence health outcomes through a variety of mechanisms, including directly intervening in buffering, or mediating effect models (Dennis 2003). The direct effect model can directly influence health outcomes by reducing social isolation and feelings of loneliness, swaying health practices and deterring maladaptive behaviors or responses, while promoting positive psychological states and individual motivation; it can also provide information, including about access to medical services, promote positive health behaviors, and prevent the risk, and progression, of physical illness while promoting recovery (Dennis 2003). Fathers felt less socially isolated because they had a specific person who was calling to talk with them, who they could more easily confide in, as the relationship was anonymous.

The moderating effect model can influence health outcomes by buffering the influence of stress on health by reducing potential for harm posed by a stressor, broadening the number of coping resources, enabling discussion of coping strategies, encouraging problem-solving techniques, inhibiting maladaptive responses, and counteracting the tendency to blame (Dennis 2003). Peer support provided in the current study had a positive impact on fathers' emotional awareness and confidence in their role as a father, and helped to foster empathy towards their partner.

The mediating effect model can indirectly influence health outcomes by increasing self-esteem through the provision of positive reinforcement, enabling vicarious experiences through role modeling and verbal persuasion (encouragement), helping to develop social comparisons that promote motivation and self-evaluation, promoting positive coping strategies, and helping participants to positively interpret emotions while encouraging cognitive restructuring (Dennis 2003). Fathers reported an increased sense of self-esteem and more confidence in their role as a father as a result of receiving peer support. They viewed the peers as role models and compared their own reaction to how the peer would have reacted in a similar situation, resulting in more positive coping strategies and more positive feelings toward their partner.

### 6.1. Adherence to the Program

Adherence is an important indicator of feasibility and refers to fathers and peers following the program's guidelines. First-time fathers and peers reported that they adhered to the program by talking on the phone at least once per month, focusing on the issues that fathers were having, and stated that peers provided support and advice. According to the peers' log, all peers provided different support, such as instrumental, emotional and informational support. By receiving these types of social supports, the father's well-being and confidence in parenting should improve (Wells et al. 2021; Hughes et al. 2020). Adhering to the program led to high participant satisfaction and some peers taking on additional fathers. Fathers felt satisfied with speaking once per month. Furthermore, fathers liked that they had the option of choosing to receive additional support, such as more phone conversations or opting to focus on personal topics during the calls. This finding is in line with previous peer support studies for mothers, which concluded that only a small number of contacts are sometimes necessary to improve clinical outcomes (Dennis et al. 2009; Dennis 2002).

*6.2. Participant Responsiveness*

As in previous telephone-based peer support for mothers (Dennis et al. 2009), the current results indicated that the TPSP-F is a desirable program for fathers. In the present study, all interviewees reported that their participation improved their ability to connect with others. While some internet-based forums, like Reddit, exist for peer support, and enable fathers to receive helpful advice and information (Teague and Shatte 2021), they can lack the connectivity that comes from developing a personal relationship with another father. Fathers communicated that they appreciated having someone who was just available for them. Similarly, Evans et al. noted that peers provided unconditional support with no judgment or pressure attached, and described the peers as being "real", and as being understanding, compassionate, and good listeners (Evans et al. 2020). Talking with friends or family may put the individual in distressing situations where they fear being judged, and strangers may therefore be a more suitable option, s this will enable them to open up and receive more objective viewpoints (Small 2017). Our study further emphasized the need for anonymity, as it allowed fathers to open up and be more vulnerable about their feelings and experiences. For example, all participants said they felt relaxed and comfortable with the telephone calls, and felt less judged than if they had physically met.

*6.3. Program Differentiation*

In this study, fathers thought that the timing of this program, either during the third trimester or soon after birth, was ideal to start receiving peer support to help them better adjust to fatherhood, which is supported by previous findings that fathers should be supported early on in the transition to fatherhood (Warren 2020). Importantly, social support helps improve health and quality of life (Gilbert et al. 2013). Fathers therefore benefit from a forum where they can discuss and receive relevant information, and also have an opportunity to engage in self-reflection (Eriksson and Salzmann-Erikson 2013). Forming social connections sooner therefore helps to mitigate negative health consequences, while promoting empowered fatherhood.

Having an open discussion, during which fathers could lead the dialogue, seemed to be acceptable to the participants. However, both fathers and peers suggested that a protocol of topics that could be discussed should be put in place, just in case there was a lull in the dialogue. As such, this program should provide all peers with a list of commonly discussed questions that can, if needed, be used to further probe fathers, if needed. In Sweden, there is a list of topics that nurses should discuss with fathers when the child is 4–6 months old, including: (i) the father–child relationship; (ii) expectations of being a parent; (iii) how the family functions (including the romantic and coparenting relationships and parental leave usage); and (iv) how the father is adjusting, which include feelings, amount (or lack) of sleep, and received or needed support (Rikshandboken 2019). However, nurses do not always follow these guidelines (Odonde et al. 2022), and fathers may therefore benefit from peers further discussing these topics.

*6.4. Frequency and Duration*

In line with other peer support programs, this study showed that receiving peer support during the first several months after birth should be as frequent as deemed necessary (by the father), and that a small number of contacts could therefore be enough (Chyzzy et al. 2020; Dennis et al. 2009). In fact, our study indicated that both fathers and peers believed that the frequency and duration of the program were sufficient and that, after only a couple of months, fathers reported feeling more secure in their new parental roles, having developed coping strategies, worked on their coparenting communication skills, and improved their overall well-being. However, a recently published literature review on peer support for maternal depression suggests that talking at least once per week is helpful for managing perinatal depression (Fang et al. 2022); in seeking to achieve clinical outcomes, the current program may therefore need to increase its frequency.

### 6.5. Peers and Mentors

Peers appreciated receiving the training on social support from mentors and considered the training when providing support to fathers. They felt relaxed when giving support and appreciated the support mentors provided when they requested further help. This is supported by previous research, which shows that brief training significantly improves peer support (Lekka et al. 2015).

### 6.6. Potential Limitations to the TPSP-F Program

To offer support to all fathers at the right time, implementing this study on a larger scale would require a larger pool of experienced fathers volunteering to be peers. As such, establishing strong collaborations with organizations that work with fathers could be helpful. Another possibility is to transition fathers to peers, so that once the program ends, the fathers can become a peer to a new prospective father. Moreover, in scaling up TPSP-F, future research should further conduct a randomized controlled trial, to show the effects of the program, compared to a control group. In building on this trial, and acknowledging the World Health Organization's call for evidence-based and peer-delivered psychological interventions in perinatal depression (World Health Organization 2015), future research should investigate if this type of program can prevent or reduce mental health problems in fathers.

### 6.7. Strengths and Limitations

A strength of our study is that we interviewed both first-time fathers and peers. By conducting this triangulation, we were able to more strongly conclude on the acceptability of the current program.

In considering the trustworthiness of the data, credibility, dependability and transferability are important considerations (Korstjens and Moser 2018). Credibility was aided by using a semi-structured interview guide, which was only slightly altered between first-time fathers and peers, meaning that each group talked about the same topics. Furthermore, first-time fathers and peers were perceived to be relaxed and talkative during the interviews, which generated rich data, resulting in two main themes respectively emerged. Dependability was attained by having two author contributors (midwifery students), who held no preconceived notions about the TPSP-F, leading the data analysis, and one program designer (removed for peer review) independently analyzing the data. The two program designers held preconceived notions (based on their previous research) that fathers do not receive enough professional support, and could benefit from peer support. They further believed that child and family outcomes are improved when fathers are more knowledgeable, well-adjusted and secure in their parental roles, and therefore wish to find ways to help fathers achieve these goals and contribute to the betterment of all members of the family. All authors agreed on the themes and categories, lending robustness to the findings.

Transferability of the findings seems relatively robust, as interviews took place with both first-time fathers and peers. Importantly, first-time fathers and peers were located anywhere in Sweden, encompassed a wide age range, could talk about any topic that was concerning the first-time father at that point in time, and were not matched with each other on the basis of any other criteria. Since both fathers and peers thought the peer support was helpful, this TPSP-F should be robustly transferable. However, we only examined the program with Swedish-speaking first-time fathers. Fathers from other backgrounds may have different expectations about fatherhood than Swedish-born fathers, who emphasize a dual-earner/dual-career model (Wells and Bergnehr 2014). In addition, multi-time fathers may have different support needs to first-time fathers. Further evidence would therefore need to be to obtained to ensure that the program is helpful for other groups of fathers.

## 7. Conclusions

Fathers need support as they transition into parenthood that will help them better adjust to their new life and changing conditions within the family. Peer support can be a

cost-effective way to receive support, and allows fathers to socially connect with others in similar situations. After qualitatively analyzing data from first-time fathers and peers who participated in a 16-week telephone-based peer support program (TPSP-F), our study found strong feasibility, adherence, and satisfaction with the program. First-time fathers and peers adhered to the program, believed the dosage/frequency was appropriate, and appreciated the anonymity of the program, resulting in positive participant responsiveness. In addition, first-time fathers and peers were satisfied with the support they received and provided, respectively. As such, peer support programs that are low-cost, such as those conducted over the telephone or on the internet, may help fathers better connect with each other, allowing new fathers to receive vital social support. Future research should collect quantitative data on multiple paternal, child and family outcomes, in order to determine the effectiveness of this type of program on the health and well-being of fathers, their offspring, and their family.

**Supplementary Materials:** The following are available online at https://www.mdpi.com/article/10.3390/socsci13030155/s1. File S1: Interview guide.

**Author Contributions:** Conceptualization, E.A.; Methodology, E.A. and M.B.W.; Validation, L.E.; Formal analysis, E.A. and M.B.W.; Investigation, E.A., L.E. and M.B.W.; Resources, E.A.; Writing—original draft, E.A., L.E. and M.B.W.; Writing—review & editing, E.A., L.E. and M.B.W.; Project administration, E.A.; Funding acquisition, E.A. All authors have read and agreed to the published version of the manuscript.

**Funding:** This research was funded by Karolinska Institutet, grant number: FS-2020:0007.

**Institutional Review Board Statement:** The study was conducted in accordance with the Declaration of Helsinki, and approved by the regional ethical board Stockholm, protocol code: 2015/1662-3115, 28 October 2015 and protocol code: 2016/217-32, 28 February 2016.

**Informed Consent Statement:** Informed consent was obtained from all subjects involved in the study.

**Data Availability Statement:** Data is unavailable due to ethical restrictions.

**Conflicts of Interest:** The authors declare no conflict of interest.

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
