# Peer review of "From Fathers to Fathers—Telephone-Based Peer Support: A Feasibility Study"

_socsci, doi:10.3390/socsci13030155_

Round 1
Reviewer 1 Report
Comments and Suggestions for Authors
The paper is very interesting. It presents the results of a qualitative study regarding the need for a support line for fathers.
The research methodology is well described and the presentation of results meets standards. The authors point out both the strengths and limitations of their study.
I found that article very useful and necessary. It is worth publishing due to the topic and a new, creative way of thinking about supporting fathers. The authors' proposal uses an already existing tool to support mothers. Testing the helpline seems obvious, but men are a specific group who do not like to show their feelings and, what is more, talk about them.
Discussing the results from the perspective of various study participants provides grounds for optimistic use of such a tool in supporting fathers. In my opinion, such a telephone line should be available in every country. Therefore, dissemination of the results is necessary.
One small suggestion - add more literature confirming the results in the Discussion section
Author Response
One small suggestion - add more literature confirming the results in the Discussion section-
Thank you for the suggestion. We have added several sources of literature to the Discussion confirming the results of our study and aiding in moving the dialogue forward now that we have our results.
Reviewer 2 Report
Comments and Suggestions for Authors
I have comments on this text. The research problem should be stated in the text and summary. Please indicate the criteria for selecting the sample for the research. There is no information about the program for fathers introduced. This is a very narrow view. The discussion and research conclusions need to be broadened. You should refer to the research of other authors and make comparisons.
Comments on the Quality of English Language-
Author Response
I have comments on this text. The research problem should be stated in the text and summary. Please indicate the criteria for selecting the sample for the research.
Thank you. We have added some information. Line 82- 95 in the background
There is no information about the program for fathers introduced.
The breakdown of how the program is run (e.g. from mentors to peers to fathers) is stated in section 4.1 of the Methods, while the content of the program and frequency of phone calls is reported in the “Frequency and Duration” section of the Methods.
This is a very narrow view. The discussion and research conclusions need to be broadened. You should refer to the research of other authors and make comparisons.
We have added in several more articles throughout the Discussion to broaden the dialogue and to help draw comparisons with previous research. We have also broaded the Conclusions section by discussing the important role peer support can play as fathers adjust to fatherhood.
Reviewer 3 Report
Comments and Suggestions for Authors
Dear Authors
Many thanks for submitting your article for peer review. Although your article is novel in the sense of examining fathers rather than mothers, the structure of the article needs serious revision in order to reach a standard necessary for publication. Please see my notes below for some detail.
Abstract
1. No need to discuss ethical approval and signing consent forms in the abstract.
2. The abstract as a whole needs to be reconfigured to make it more succinct and flow better. At present, it reads choppy and has material in it that are not necessary. More emphasis on the themes and sub themes created is required.
Keywords - If of equal value, place them in alphabetical order.
Background - The information provided in the entire introduction to your study is sparse at best. There needs to be more critical discussion of the current evidence base. I agree that this is minimal when it comes to fathers but you need to give a more extensive introduction touching on the possibilities as to why research on fathers is lacking. In addition, you could provide a paragraph on peer support and peer support groups and then tie in together at the end through study aim and objectives.
Please refrain from using i.e, etc, e.g in academic writing.
Methods - As a whole, the methods looks disjointed and for a while I thought some of the earlier content may have been useful for the introduction. Your methods need to be succinct and follow headings such as study design, methodological orientation, data collection, recruitment, ethical approval and data analysis. Also explain your rationale for doing a qualitative study rather than a mixed method or quantitative study and also include your ontological and epistemological stance in this research and make reference in the discussion as to how this shaped your study.
Results - Again, it is very disjointed and not very well presented. You need to first tell us the themes and then maybe a table with the themes and sub themes. At the moment, this information is hard to grasp. Additionally, the quotations need to be made distinct from the remainder of the text. Your results are impressive, so make them look impressive on the manuscript. Review the entire findings/results so that it is crystal clear what your results are and make sure it flows with on leading to another. To do this, you may need to review you selection of themes and sub-themes again. In addition, you separated fathers from peers and I am not sure if this is the correct move for the paper. A more consolidated, succinct and clear findings is required.
Discussion - Ensure you focus on the lack of data on fathers and how this is novel in that approach. Discuss the limitations of qualitative research such as the inability to generalise findings. Add more empirical evidence to the discussions to contextualise the findings to the current literature base. Another programme of work that may be useful to review as part of your introduction and discussion is the PRIMERA project as this clearly stipulates the lack of focus on fathers.
All in all, this paper has massive potential given the fact that it focuses on fathers rather than mothers which is novel. However, you need to improve your presentation of the entire manuscript and contextualise it more on the current literature base. Keep going, you will get there. I look forward to reviewing the revised manuscript.
Comments on the Quality of English LanguageQuality of English is ok, could use slight improvement as noted earlier.
Author Response
Abstract
- No need to discuss ethical approval and signing consent forms in the abstract.
Thank you. We have now removed this sentence from the abstract.
- The abstract as a whole needs to be reconfigured to make it more succinct and flow better. At present, it reads choppy and has material in it that are not necessary. More emphasis on the themes and sub themes created is required.
Thank you for the suggestion. We have re-written the abstract to be less choppy, and added in more sentences to the results.
Keywords - If of equal value, place them in alphabetical order.
The keywords are now all in alphabetical order.
Background - The information provided in the entire introduction to your study is sparse at best. There needs to be more critical discussion of the current evidence base. I agree that this is minimal when it comes to fathers but you need to give a more extensive introduction touching on the possibilities as to why research on fathers is lacking. In addition, you could provide a paragraph on peer support and peer support groups and then tie in together at the end through study aim and objectives.
Thank you for the suggestions. We have now added a whole section on peer support, both theoretically why it’s important, how it’s defined, and that interventions are missing, but of some of those that exist, we summed their primary findings.
Please refrain from using i.e, etc, e.g in academic writing.
These terms have all been removed from the manuscript.
Methods - As a whole, the methods looks disjointed and for a while I thought some of the earlier content may have been useful for the introduction. Your methods need to be succinct and follow headings such as study design, methodological orientation, data collection, recruitment, ethical approval and data analysis. Also explain your rationale for doing a qualitative study rather than a mixed method or quantitative study and also include your ontological and epistemological stance in this research and make reference in the discussion as to how this shaped your study.
Thank you for the suggestions. We have motivated our method at the beginning of the Methods section by saying:
“The current study used a qualitative design, as it highlights the significance of grasping the meanings that individuals attach to their experiences. This methodology acknowledges reality’s subjectivity and underscores the need to comprehend interpretations from individuals within these distinct contexts. This process involves recognizing how fathers and peers perceive and interpret the same phenomenon, acknowledging the variations among individuals. Additionally, using a qualitative approach can provide insights for developing interventions and can assist in identifying the obstacles and catalysts for successful implementation.”
We further have explained our stance in this research by stating in the Discussion
“The two program designers held preconceived notions based on their previous research that fathers do not receive enough professional support, and may benefit from peer support. They further hold beliefs that child and family outcomes are improved when fathers are more knowledgeable, well-adjusted and secure in their parental roles; thus, they wish to find ways to help fathers achieve these goals for the betterment of all members of the family.” Line 614-619
Results - Again, it is very disjointed and not very well presented. You need to first tell us the themes and then maybe a table with the themes and sub themes. At the moment, this information is hard to grasp. Additionally, the quotations need to be made distinct from the remainder of the text. Your results are impressive, so make them look impressive on the manuscript. Review the entire findings/results so that it is crystal clear what your results are and make sure it flows with on leading to another. To do this, you may need to review you selection of themes and sub-themes again. In addition, you separated fathers from peers and I am not sure if this is the correct move for the paper. A more consolidated, succinct and clear findings is required.
Thank you for the suggestions. We agree, and we thought we had tables added in the first submission, but something clearly happened where they didn’t show up. To help make the presentation of the results less disjointed, we have re-included those tables of the themes and categories, as well as report the names (small changes) of the themes in the opening sentences of the Results. Line 191-194.
To help make the quotations distinct, we have moved the margins, so that the quotes stand out more clearly.
We discussed the possibility of combining the fathers’ and peers’ findings to report them together. However, combining the results would lead to losing important details (similarities and differences between the two groups of interviews). We feel these details are important since very little research has been done on fathers’ and peers’ views of participating in a peer support program. However, throughout the Discussion section, we combine the findings from fathers and peers when explaining and translating the findings, and thus make the primary findings more succinct and within the context of the larger field.
Discussion - Ensure you focus on the lack of data on fathers and how this is novel in that approach. Discuss the limitations of qualitative research such as the inability to generalise findings. Add more empirical evidence to the discussions to contextualise the findings to the current literature base. Another programme of work that may be useful to review as part of your introduction and discussion is the PRIMERA project as this clearly stipulates the lack of focus on fathers.
Rather than discussing the generalizability of the findings, which is common in quantitative research, we have discussed the potential transferability of the findings. In fact, there is now an entire paragraph in the Strengths and Limitations section going over the transferability of the current findings. Line 164
More empirical evidence has been added to the Discussion throughout the whole discussion section, including multiple paragraphs added in immediately below the first paragraph to place our findings in a larger context. Begin in Line 479.
Round 2
Reviewer 3 Report
Comments and Suggestions for Authors
Dear Authors
Many thanks for resubmitting this paper for review. You have addressed all my comments adequately and I have no problem recommending the paper for publication. Wishing you and this paper all the best.